# Activity-dependent regulation of T-type calcium channels by submembrane calcium ions

Magali Cazade[1,2], Isabelle Bidaud[1,2], Philippe Lory[1,2]*, Jean Chemin[1,2]*

[1]IGF, CNRS, INSERM, University of Montpellier, Montpellier, France; [2]LabEx 'Ion Channel Science and Therapeutics', Montpellier, France

**Abstract** Voltage-gated $Ca^{2+}$ channels are involved in numerous physiological functions and various mechanisms finely tune their activity, including the $Ca^{2+}$ ion itself. This is well exemplified by the $Ca^{2+}$-dependent inactivation of L-type $Ca^{2+}$ channels, whose alteration contributes to the dramatic disease Timothy Syndrome. For T-type $Ca^{2+}$ channels, a long-held view is that they are not regulated by intracellular $Ca^{2+}$. Here we challenge this notion by using dedicated electrophysiological protocols on both native and expressed T-type $Ca^{2+}$ channels. We demonstrate that a rise in submembrane $Ca^{2+}$ induces a large decrease in T-type current amplitude due to a hyperpolarizing shift in the steady-state inactivation. Activation of most representative $Ca^{2+}$-permeable ionotropic receptors similarly regulate T-type current properties. Altogether, our data clearly establish that $Ca^{2+}$ entry exerts a feedback control on T-type channel activity, by modulating the channel availability, a mechanism that critically links cellular properties of T-type $Ca^{2+}$ channels to their physiological roles.

*For correspondence: philippe.
lory@igf.cnrs.fr (PL); jean.chemin@
igf.cnrs.fr (JC)

**Competing interests:** The
authors declare that no
competing interests exist.

**Reviewing editor:** Baron
Chanda, University of Wisconsin-
Madison, United States

## Introduction

Voltage-gated $Ca^{2+}$ channels (VGCCs) are unique among voltage-gated ion channels because the permeant $Ca^{2+}$ ion also acts as an intracellular second messenger, triggering diverse cellular functions (*Berridge et al., 2003*). VGCCs are therefore involved in neuronal and cardiac excitability as well as in muscle contraction, neurotransmitter release, hormone secretion and gene expression (*Berridge et al., 2003*; *Mangoni and Nargeot, 2008*; *Catterall, 2011*; *Simms and Zamponi, 2014*; *Zamponi et al., 2015*). Consequently the modulation of VGCC activity plays a pivotal role in the regulation of cardiac and brain activities and this modulation is controlled by a variety of processes, including intracellular $Ca^{2+}$ itself, which provides an important $Ca^{2+}$-driven feedback control (*Eckert and Chad, 1984*; *Zühlke et al., 1999*; *Peterson et al., 1999*; *Liang et al., 2003*; *Green et al., 2007*; *Tsuruta et al., 2009*; *Oliveria et al., 2012*; *Hall et al., 2013*; *Zamponi et al., 2015*).

VGCCs comprise three distinct subfamilies classified with respect to their biophysical and pharmacological (type), and molecular (Cav) entities: the L-type / Cav1, the N-, P/Q-, R-type / Cav2 and the T-type / Cav3 channels (*Ertel et al., 2000*). It was well demonstrated that both Cav1 and Cav2 channels are modulated by intracellular $Ca^{2+}$ (*Liang et al., 2003*; *Dick et al., 2008*). For the Cav1 / L-type VGCCs, this $Ca^{2+}$ feedback mechanism has been extensively studied in a wide spectrum of biological contexts and a rise in submembrane $Ca^{2+}$ concentration induces complex effects depending on both the $Ca^{2+}$ concentration and the duration of the $Ca^{2+}$ entry (*Eckert and Chad, 1984*; *Zühlke et al., 1999*; *Peterson et al., 1999*; *Liang et al., 2003*; *Green et al., 2007*; *Tsuruta et al., 2009*; *Oliveria et al., 2012*; *Hall et al., 2013*). At the millisecond time scale, the $Ca^{2+}$ entry via L-type channels induces a $Ca^{2+}$-dependent inactivation (CDI) characterized by an acceleration of

**eLife digest** Neurons, muscle cells and many other types of cells use electrical signals to exchange information and coordinate their behavior. Proteins known as calcium channels sit in the membrane that surrounds the cell and can generate electrical signals by allowing calcium ions to cross the membrane and enter the cell during electrical activities. Although calcium ions are needed to generate these electrical signals, and for many other processes in cells, if the levels of calcium ions inside cells become too high they can be harmful and cause disease.

Cells have a "feedback" mechanism that prevents calcium ion levels from becoming too high. This mechanism relies on the calcium ions that are already in the cell being able to close the calcium channels. This feedback mechanism has been extensively studied in two types of calcium channel, but it is not known whether a third group of channels – known as Cav3 channels – are also regulated in this way.

Cav3 channels are important in electrical signaling in neurons and have been linked with epilepsy, chronic pain and various other conditions in humans. Cazade et al. investigated whether calcium ions can regulate the activity of human Cav3 channels. The experiments show that these channels are indeed regulated by calcium ions, but using a distinct mechanism to other types of calcium channels. For the Cav3 channels, calcium ions alter the gating properties of the channels so that they are less easily activated . As a result, fewer Cav3 channels are "available" to provide calcium ions with a route into the cell.

The next steps following on from this work will be to identify the molecular mechanisms underlying this new feedback mechanism. Another challenge will be to find out what role this calcium ion-driven feedback plays in neurological disorders that are linked with altered Cav3 channel activity.

their inactivation kinetics (*Eckert and Chad, 1984*; *Zühlke et al., 1999*; *Peterson et al., 1999*; *Liang et al., 2003*; *Hall et al., 2013*). For several seconds to a few minutes of stimulation, the cumulative $Ca^{2+}$ entry induces a decrease of the L-type current amplitude, which is reversible if stimulation ceases for several minutes (*Eckert and Chad, 1984*; *Oliveria et al., 2007*, *2012*; *Hall et al., 2013*). For longer period of stimulation, or activation of the ionotropic NMDA receptors, L-type channels are internalized, potentially degraded in lysosomes or recycled to the plasma membrane depending on the amount and the duration of the $Ca^{2+}$ entry (*Green et al., 2007*; *Tsuruta et al., 2009*; *Hall et al., 2013*). This precise $Ca^{2+}$-dependent regulation of the L-type channel activity has a strong physiological role in avoiding cytotoxicity arising from $Ca^{2+}$ overload (*Lee et al., 1999*; *Berridge et al., 2003*; *Green et al., 2007*; *Tsuruta et al., 2009*; *Hall et al., 2013*). Consequently, alteration of the $Ca^{2+}$-dependent regulation of the L-type channels is deleterious and has important pathophysiological consequences as observed in the Timothy syndrome (*Splawski et al., 2004*; *Barrett and Tsien, 2008*; *Blaich et al., 2012*; *Limpitikul et al., 2014*; *Dick et al., 2016*).

Contrasting with this well-established $Ca^{2+}$-dependent regulation of the L-type channels, it is presently unknown whether a change in intracellular $Ca^{2+}$ concentration is involved in regulating T-type $Ca^{2+}$ channel activity. The low-voltage-activated, T-type/Cav3 channels are specifically activated by small membrane depolarization below the threshold of classical sodium action potentials, producing a $Ca^{2+}$ entry near the resting membrane potential and low-threshold $Ca^{2+}$ spikes (*Huguenard, 1996*, *1998*; *Perez-Reyes, 2003*; *Zamponi, 2016*). Importantly, availability of Cav3 channels is critically regulated by the resting membrane potential to control T-type channel activity. Because of the negative range of their steady-state inactivation ($V_{0.5}$ near $-70$ mV), Cav3 channels are partially inactivated (reduced availability) in the range of the resting membrane potential of most neurons and a membrane hyperpolarization (usually triggered by inhibitory postsynaptic events) is needed to allow their recovery from inactivation (de-inactivation) and their subsequent opening (*Huguenard, 1996*; *Perez-Reyes, 2003*; *Zamponi, 2016*). This behavior is of particular importance in many types of neurons, in which Cav3 channels mediate rebound burst firing, especially in the thalamo-cortical circuit, where Cav3 channels control transition between awake and sleep states (*Huguenard, 1996*; *Perez-Reyes, 2003*; *Beenhakker and Huguenard, 2009*; *Zamponi, 2016*;

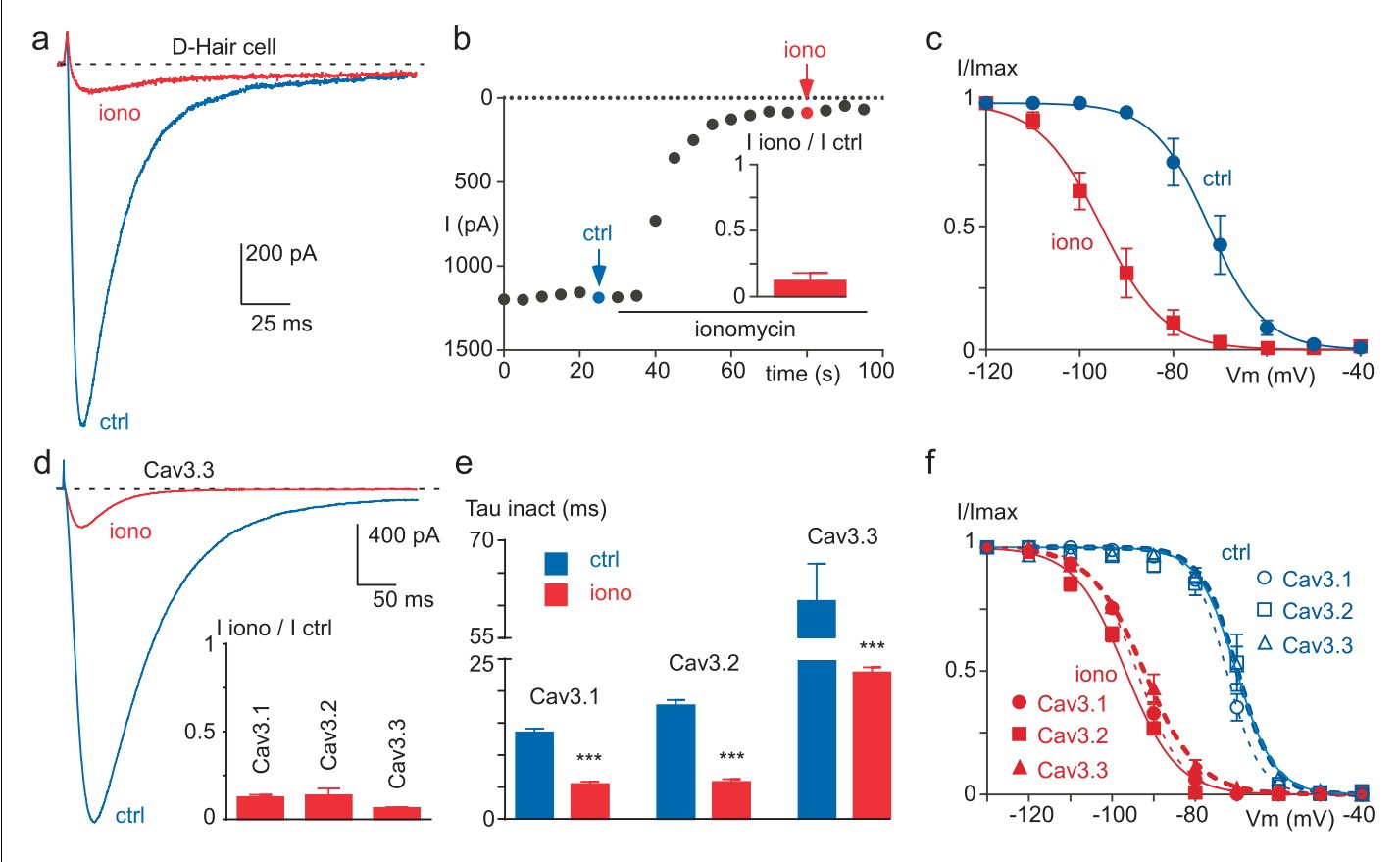

**Figure 1.** Ionomycin induces inhibition of neuronal T-type and recombinant Cav3 currents. (a–c) Extracellular application of ionomycin induces inhibition of the native T-type current in D-hair sensory neurons. (a) Effect of 10 µM ionomycin (iono) on the T-type current recorded from a D-hair sensory neuron. (b) Time course and average effect (inset) of ionomycin (n = 5). (c) Steady-state inactivation of the native T-type current before (ctrl) and after ionomycin application (n = 5). (d–f) Ionomycin induces inhibition of the recombinant Cav3.1, Cav3.2 and Cav3.3 currents. (d) Effect of ionomycin application (10 µM) on the Cav3.3 current amplitude. Average effect of ionomycin on Cav3.1, Cav3.2 and Cav3.3 currents (inset, n = 6–7 per bar). (e) Inactivation kinetics of the Cav3 currents in the absence and in the presence of ionomycin (n = 6–7 per bar). (f) Steady-state inactivation of Cav3.1, Cav3.2 and Cav3.3 currents before and after ionomycin application (n = 6–7). In (a), (b), (d) and (e), the currents were recorded at −30 mV from a holding potential (HP) of −80 mV. In (c) and (f), the currents were elicited at −30 mV from HPs ranged from −130 to −40 mV (5 s duration) and the data were fitted with the Boltzmann equation.

*Tatsuki et al., 2016*). Overall, Cav3 channels are involved in the control of the $Ca^{2+}$ homeostasis (*Chemin et al., 2000*; *Bijlenga et al., 2000*; *Perez-Reyes, 2003*), in $Ca^{2+}$-dependent differentiation of neuronal, muscular and neuroendocrine cells (*Bijlenga et al., 2000*; *Mariot et al., 2002*; *Chemin et al., 2002b*), as well as in $Ca^{2+}$ overload toxicity in ischemia (*Nikonenko et al., 2005*; *Bancila et al., 2011*; *Gouriou et al., 2013*). Importantly, an increase in the Cav3 channel activity has been implicated in several diseases including epilepsy, chronic pain, autism and primary aldosteronism (*Beenhakker and Huguenard, 2009*; *Zamponi, 2016*). Although it is evident that a tight control of Cav3 channel activity is necessary to maintain $Ca^{2+}$ homeostasis, there is no evidence yet that Cav3 channels are regulated by intracellular $Ca^{2+}$ ions and/or by $Ca^{2+}$ entry.

In this study, we have designed complementary electrophysiological experiments to explore whether the T-type/Cav3 channels are modulated by intracellular $Ca^{2+}$ concentration. We document a feedback control mechanism that relies on $Ca^{2+}$ entry via activated Cav3 channels or nearby $Ca^{2+}$-permeable receptors. We provide evidence that dynamic changes and localized increase in the intracellular $Ca^{2+}$ concentration at the vicinity Cav3 channels control availability of these channels, which underlies this novel regulation.

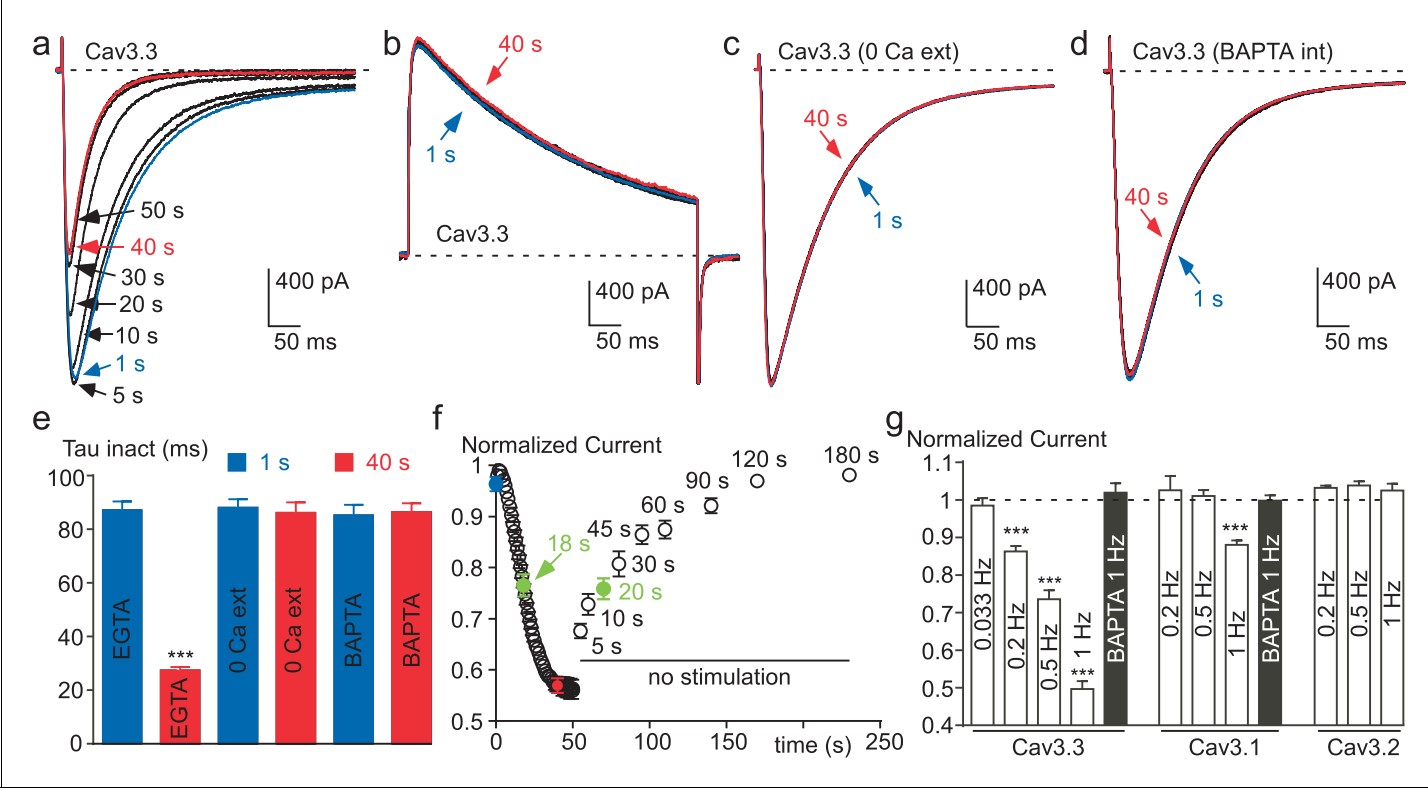

**Figure 2.** $Ca^{2+}$ induces a time-dependent inhibition of the Cav3.3 current at high frequency of stimulation. (**a**) Typical examples of Cav3.3 currents elicited using a 1 Hz test pulse (TP) stimulation of −30 mV (450 ms duration) from a HP of −100 mV. The traces obtained at the beginning of the stimulation (1 s) and after 40 s stimulation are indicated in blue and red, respectively. (**b–d**) Similar experiments for a TP of +100 mV (**b**), in the absence extracellular $Ca^{2+}$ (**c**, TP −30 mV) and in the presence of intracellular BAPTA (**d**, TP −30 mV). (**e**) Inactivation kinetics of the Cav3.3 current measured at the beginning (1 s, blue bars) and after 40 s stimulation (red bars, n = 15–28 per bar). (**f**) Time-course of the Cav3.3 current inhibition during 1 Hz stimulation and time-course of the recovery of the Cav3.3 current as a function of time after stimulation (n = 13–28 per point). The half-time of both inhibition and recovery of the Cav3.3 current are indicated in green. (**g**) Summary of the data obtained on the three Cav3 currents at different frequencies of TP stimulation (n = 5–40 per bar).

The following figure supplements are available for figure 2:

**Figure supplement 1.** The effect of free $Ca^{2+}$ application on the Cav3.3 current recorded in cell-free inside-out patches.

**Figure supplement 2.** Inhibition of the Cav3.1 current at a high frequency of TP stimulation but not of the Cav3.2 current.

# Results

## Inhibition of the native T-type and the recombinant Cav3 currents by ionomycin

In order to evaluate whether T-type channels would be regulated by a rise in intracellular $Ca^{2+}$ ($[Ca^{2+}]_{INT}$), we first used the $Ca^{2+}$ ionophore ionomycin. In D-hair mechanoreceptor sensory neurons, which specifically express a high density of T-type channels (*Shin et al., 2003*; *Dubreuil et al., 2004*; *Voisin et al., 2016*), extracellular perfusion of 10 µM ionomycin induced a potent decrease of the T-type current (*Figure 1a*). This current inhibition (~85% in average) occurred in the minute range (*Figure 1b*) and was associated with a large hyperpolarizing shift in the steady-state inactivation curve (~22 mV, p<0.001, *Figure 1c*). Similar findings were obtained with the three cloned Cav3 channels transiently expressed in tsA-201 cells. Ionomycin potently inhibited the Cav3.1, Cav3.2 and Cav3.3 currents by about 80% (*Figure 1d*). This effect was combined with an acceleration of the inactivation kinetics (~3 times, *Figure 1e*) and a hyperpolarizing shift in the steady-state inactivation curve (~23 mV, p<0.001, *Figure 1f*).

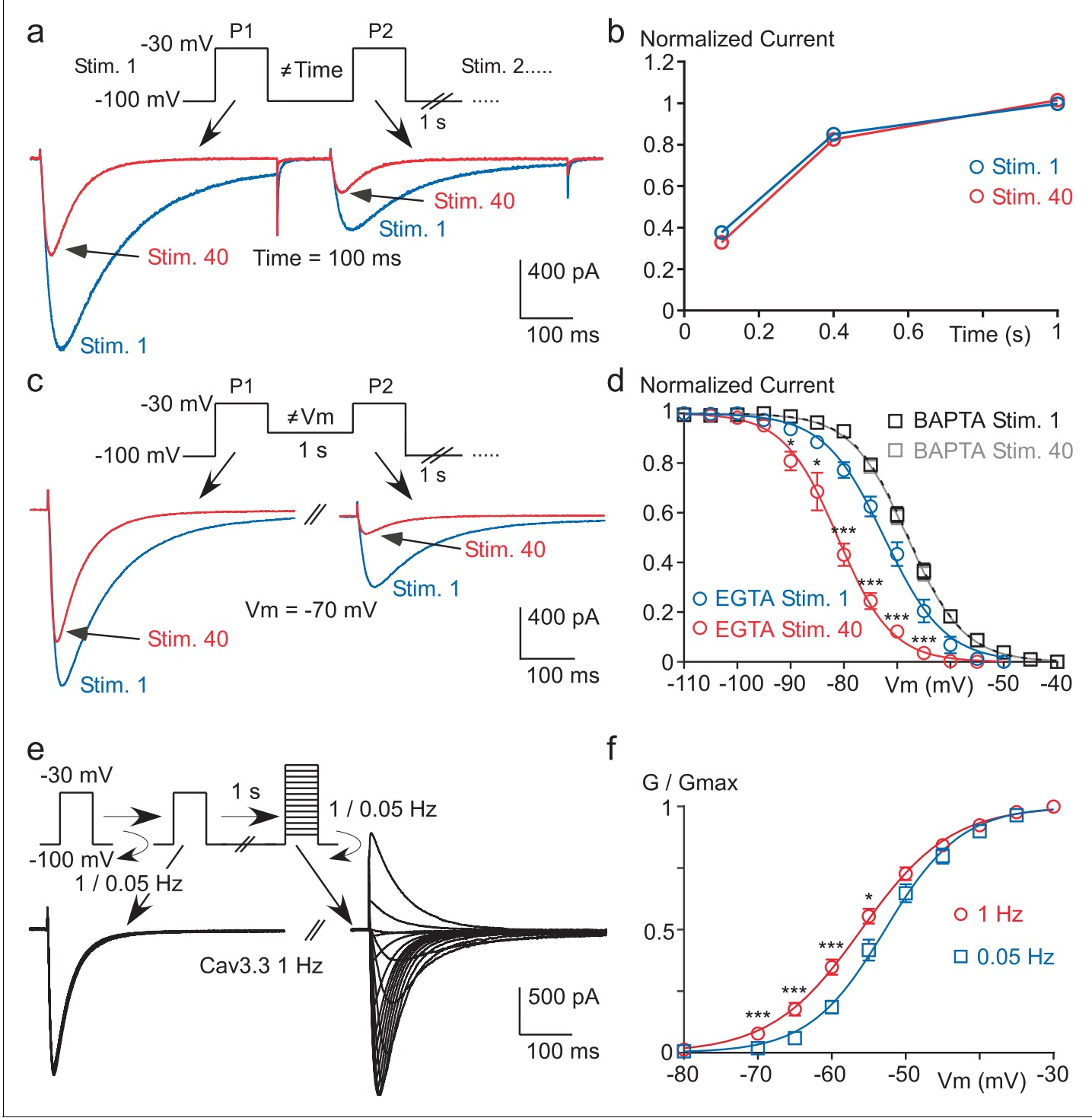

**Figure 3.** High frequency stimulation induces a Ca$^{2+}$-dependent negative shift in the Cav3.3 steady-state inactivation properties. (a–b) Recovery from short-term inactivation of the Cav3.3 current measured by a paired-pulse stimulation (inter-stimulation 1 s) applied 40 times (Stim. 1 to Stim. 40). The interval between the first pulse (P1) and the second pulse (P2), i.e. interpulse, is 100, 400 or 1000 ms, as presented in (a) for an interpulse interval of 100 ms. The recovery from short-term inactivation (P2/P1), as a function of the interpulse duration is quantified for the first stimulation (Stim. 1) and the 40th stimulation (Stim. 40) (b, n = 5–7 per point). (c–d) Steady-state inactivation of the Cav3.3 current measured using a paired-pulse stimulation applied 40 times. The Vm between the two pulses ranged from −110 to −40 mV, as illustrated in (c) for a Vm of −70 mV. Steady-state inactivation (measured at P2) as a function of the Vm is determined for the first stimulation (Stim. 1) and for the 40th stimulation (Stim. 40) (d, n = 5–9 per point). (e–f) Current-voltage (I–V) protocol (e) and activation curve (f) of the Cav3.3 current during 1 Hz or 0.05 Hz stimulation. The Cav3.3 current was stimulated at 1 Hz or 0.05 Hz

*Figure 3 continued on next page*

*Figure 3 continued*

until reaching the steady-state just before I-V protocols, which were performed by a double-pulse protocol to maintain the 1 Hz stimulation effect (n = 17).

The following figure supplement is available for figure 3:

**Figure supplement 1.** Paired-pulse stimulation at high frequency induces a negative shift in the Cav3.1 steady-state inactivation properties.

## Inhibition of Cav3.3 current at high frequency of stimulation is caused by the $Ca^{2+}$ entry

Considering that recombinant Cav3 channels, especially Cav3.3 channels, can generate large entry of $Ca^{2+}$ during fast stimulation protocols (*Huguenard, 1998*; *Kozlov et al., 1999*; *Chemin et al., 2002a*; *Perez-Reyes, 2003*), we have investigated whether these channels might be modulated by their own activity. Cav3 currents were recorded in the presence of 2 mM extracellular $Ca^{2+}$ using fast test-pulse (TP) stimulation (1 Hz), which allows a cumulative $Ca^{2+}$ entry. Experiments were performed after dialyzing the cell with 10 mM EGTA, which delimits the change in $Ca^{2+}$ concentration only at the vicinity of the channel (*Marty and Neher, 1985*; *Roberts, 1993*; *Deisseroth et al., 1996*). The Cav3.3 current amplitude recorded using fast TP stimulation (1 Hz) progressively decreased to ~50% of the control value (*Figure 2a*). In average, the current decrease was maximal and stable after ~40 s with an average half-time of 18 s (*Figure 2a and f*). The current decrease was also associated with a marked acceleration of the inactivation kinetics (*Figure 2a*), ~3 times after 40 s stimulation (*Figure 2e*). This effect was fully reversible and activity-dependent since the stopping of the TP stimulation for only 5 s already induced ~30% recovery of the current. Recovery was complete after ~2 min without TP stimulation with an average half-time of 20 s (*Figure 2f*).

In contrast with the data obtained for a TP at −30 mV (*Figure 2a*), the Cav3.3 current was unchanged when the TP was set at +100 mV (a membrane potential value above the reversal potential of Cav3 current, leading to an outward current, *Figure 2b*). These experiments clearly indicate that the decrease in current amplitude observed for a fast TP stimulation at −30 mV does not involve a voltage-dependent inactivation process that would occur at high frequency of stimulation, but is rather related to the $Ca^{2+}$ entry via Cav3.3 channels. Also, in the absence of extracellular $Ca^{2+}$, the sodium inward current through Cav3.3 channels remained unchanged during the time of the fast TP protocol at −30 mV (*Figure 2c*). In addition, no change in Cav3.3 current properties were obtained in the presence of 2 mM extracellular $Ca^2$ when cells were dialyzed with an intracellular medium containing BAPTA instead of EGTA (*Figure 2d and e*). This difference in susceptibility to BAPTA and EGTA is characteristic of a process driven by a localized rise in submembrane $Ca^{2+}$, without the need for a global $Ca^{2+}$ increase (*Marty and Neher, 1985*; *Roberts, 1993*; *Deisseroth et al., 1996*).

We further investigated whether $Ca^{2+}$ ions could affect the Cav3.3 current in cell-free inside-out patches (*Figure 2—figure supplement 1*). Cav3.3 currents were recorded by voltage-ramps in the presence of 100 mM external $Ba^{2+}$ whereas $Ca^{2+}$-containing solutions were applied to the internal side of the membrane in the inside-out patch configuration (*Figure 2—figure supplement 1a*). Because of the surface charge effect due to the use of 100 mM $Ba^{2+}$, voltage ramps were applied from a HP −50 mV to match whole-cell experiments (see also *Figure 4*). In this configuration, the application of 1, 10 or 100 µM $Ca^{2+}$ during more than 60 s (*Figure 2—figure supplement 1b*) did not induce a significant inhibition of the Cav3.3 current as compared to a control solution containing 1 mM EGTA/0 mM $Ca^{2+}$ (*Figure 2—figure supplement 1c*), suggesting that $Ca^{2+}$-induced Cav3.3 inhibition requires some additional components preserved in the whole-cell configuration. However, our data do not exclude a possible direct effect of $Ca^{2+}$ ions at higher (mM) concentrations.

The decrease in Cav3.3 current gradually developed with the increase in TP frequency. While no decrease was observed at low frequency of TP (0.033 Hz), the decrease in Cav3.3 current became significant at 0.2 Hz and further increased at 0.5 and 1 Hz (*Figure 2g*). Similar experiments were conducted with the Cav3.1 and Cav3.2 T-type channels. The amplitude of Cav3.1 current decreased modestly and only at the TP frequency of 1 Hz (*Figure 2g* and *Figure 2—figure supplement 2a–b*). Contrasting with the results described above, the Cav3.2 current showed no inhibition but rather a small increase in amplitude at fast stimulation (*Figure 2g* and *Figure 2—figure supplement 2d–e*).

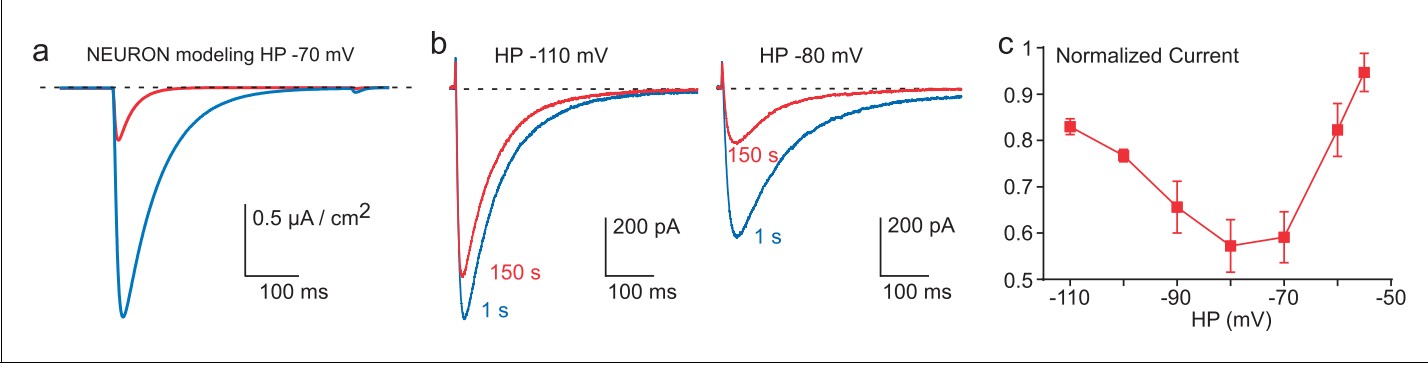

**Figure 4.** Inhibition of Cav3.3 current as a function of the HP. (a) Simulation of the Cav3.3 current at −30 mV from an HP of −70 mV. The Cav3.3 current was modeled from experiments presented in *Figure 3*. The blue trace corresponds to the Cav3.3 current properties obtained before the 1 Hz stimulation whereas the red trace corresponds to the Cav3.3 current properties obtained after 40 s stimulation at 1 Hz. (b) Cav3.3 current elicited at a frequency of 0.2 Hz from a HP of −110 mV (left panel) and a HP of −80 mV (right panel). (c) Inhibition of Cav3.3 current after 150 s stimulation as a function of the HP (n = 5–8 per point).

For Cav3.1 channels, the decrease in current amplitude was associated with faster inactivation kinetics (*Figure 2—figure supplement 2c*), while inactivation kinetics of the Cav3.2 current was unchanged (*Figure 2—figure supplement 2f*). Similar to that described for the Cav3.3 current, Cav3.1 current amplitude and inactivation kinetics were unchanged after dialyzing the cells with BAPTA (*Figure 2g*).

## Ca$^{2+}$ entry during fast TP stimulation induces a negative shift in the steady-state inactivation

We next investigated the biophysical mechanism underlying the Cav3.3 current decrease. We first hypothesized that the recovery from inactivation of the Cav3.3 current might be affected during a fast stimulation protocol in a Ca$^{2+}$-sensitive manner. A paired-pulse protocol with increasing inter-pulse durations (100, 400 or 1000 ms) was designed to analyze the kinetics of recovery from the first (Stim 1) to the fortieth (Stim 40) paired-pulse stimulation, as exemplified for an inter-pulse duration of 100 ms in *Figure 3a*. These experiments revealed that the recovery kinetics of the Cav3.3 current was unaffected at the three inter-pulse durations tested (*Figure 3b*). In contrast, we found that the steady-state inactivation of Cav3.3 current was strongly modified using fast stimulation protocols. This was evidenced using a paired-pulse protocol with variable inter-pulse potentials ranging from −40 to −110 mV (as exemplified for an inter-pulse potential of −70 mV in *Figure 3c*). This fast stimulation protocol produced a ~10 mV hyperpolarizing shift in the $V_{0.5}$ value of the steady-state inactivation curve (from −72.4 mV for Stim 1 to −81.1 mV for the Stim 40, p<0.001), without any change in the slope of the inactivation curve (*Figure 3d*). Importantly, this effect was lost in the presence of intracellular BAPTA (*Figure 3d*). In addition, we found that the fast stimulation of Cav3.3 current also induced a small but significant leftward shift in the steady-state activation (from −52.6 to −55.9 mV, p<0.01) as well as an increase in the slope of the activation curve (from 4.7 to 5.6 mV, p<0.05, *Figure 3e–f*).

Similar findings were obtained with Cav3.1 channels since these paired-pulse protocols revealed a ~5 mV hyperpolarizing shift of the steady-state inactivation curve (p<0.001, *Figure 3—figure supplement 1a–b*) with no significant effect on the recovery kinetics (*Figure 3—figure supplement 1c*) and the steady-state activation curve of the Cav3.1 current (*Figure 3—figure supplement 1d*).

## Ca$^{2+}$-dependent inhibition of T-type channels is higher at physiological resting membrane potentials

These latter findings strongly suggest that the decrease in Cav3.3 current observed in fast stimulation protocols might be more important in the range of physiological resting membrane potentials ~−70 / −80 mV, for which Cav3.3 channels are partially inactivated. In order to test this possibility, the Cav3.3 current properties before and after fast stimulation were modeled using the Hodgkin-

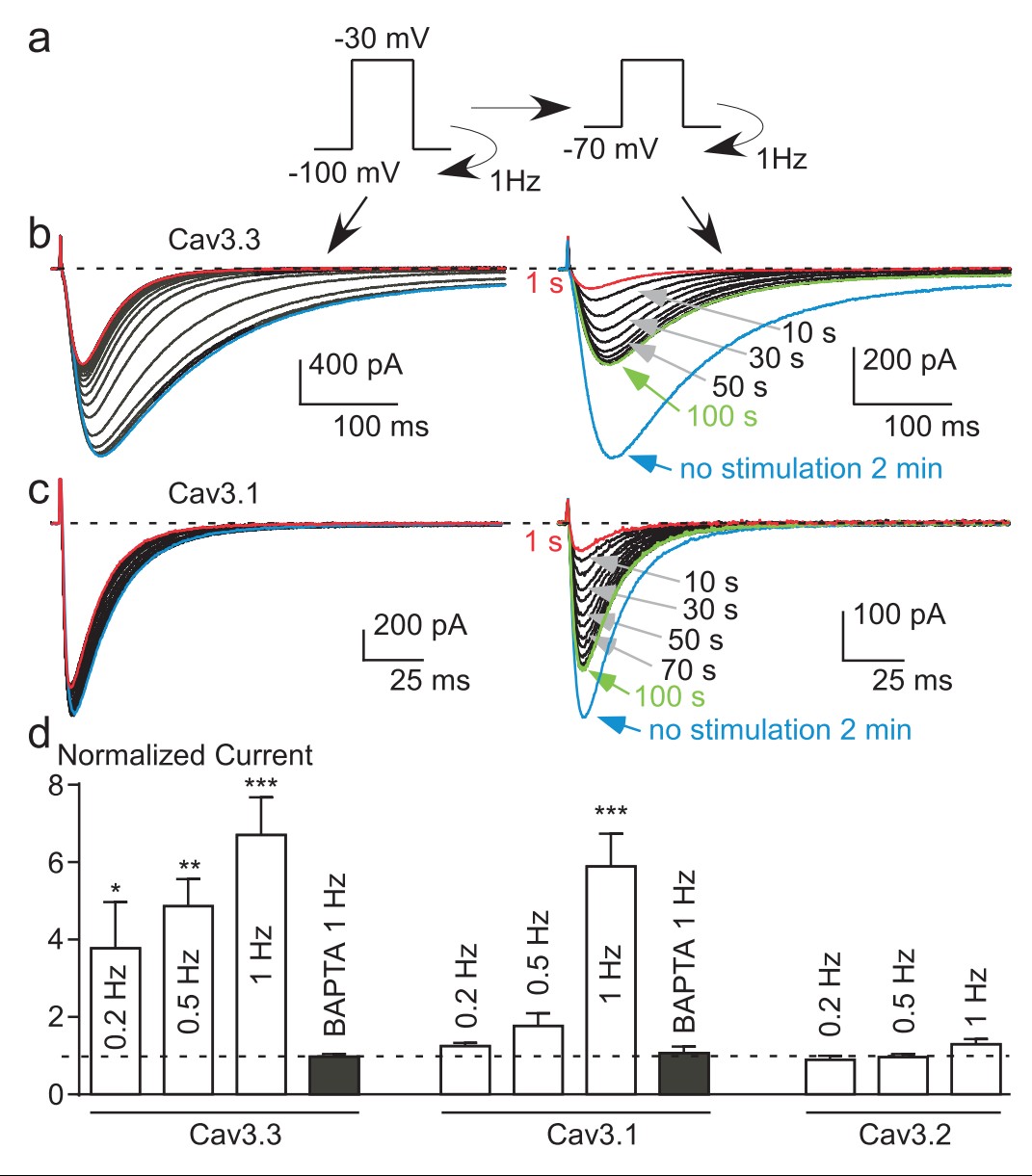

**Figure 5.** Potent T-type current recovery at physiological resting potentials. (**a–c**) Fast TP stimulation (TP −30 mV) using HP of −100 mV to induce inhibition, switched to a HP of −70 mV to induce recovery (see protocol in (**a**)). The inhibition of Cav3.3 and Cav3.1 currents was measured using fast TP stimulation at a HP of −100 mV (**b–c**, left panels) whereas the recovery of the Cav3.3 and Cav3.1 currents was measured on a HP of −70 mV (**b–c**, right panels). (**d**) Quantification of the increase (recovery) in Cav3.3, Cav3.1 and Cav3.2 currents (at HP −70 mV) as a function of the TP stimulation frequency (n = 5–40 per bar). The normalized current corresponds to the ratio of the current obtained after 2 min without stimulation (blue trace) to the initial current (1 s, red trace) recorded at a HP −70 mV.

The following figure supplement is available for figure 5:

**Figure supplement 1.** Modulation of the Cav3.2 Met1549Ile mutant channel at a high frequency of stimulation.

Huxley formalism as previously described for the native T-type current in thalamic neuron (*Huguenard and Prince, 1992*; *Destexhe et al., 1996*). We used the NEURON simulation environment (*Hines and Carnevale, 1997*) as modified previously in order to perform voltage-clamp

experiments (*Destexhe et al., 1996*). In this simulation, the model indicated that the Cav3.3 current elicited from HP −70 mV would decrease ~75% (*Figure 4a*) according to the shift in the steady-state inactivation observed in *Figure 3d*. To validate experimentally these data, we performed voltage-clamp experiments at different HPs. Using 0.2 Hz frequency of TP stimulation (to allow current recovery at more depolarized HPs), the decrease in Cav3.3 current was below 20% at HP −110 mV whereas the current decrease was more prominent (~40%, p<0.01) at HP −80 mV (*Figure 4b*). Interestingly, the decrease in Cav3.3 current was also reduced at HPs above −70 mV and was less than 10% at HP −55 mV (*Figure 4c*). This U-shaped relationship (*Figure 4c*) could be explained by two interlinked mechanisms: (i) at negative membrane potentials (below −90 mV) the shift in the steady-state inactivation curve has little impact on the decrease in current amplitude (see *Figure 3d*); (ii) the small decrease observed at HPs above −70 mV would be related to the reduced $Ca^{2+}$ entry at depolarized membrane potentials.

To directly test this latter hypothesis, the Cav3.3 current was recorded using fast TP stimulation (1 Hz) at HP −100 mV and then immediately at HP −70 mV (*Figure 5a*). At HP −100 mV during 40 s, the fast TP stimulation induced a large $Ca^{2+}$ entry as evidenced by the decrease in Cav3.3 current amplitude (*Figure 5b*, left panel). Then, when the HP was immediately switched to −70 mV (*Figure 5b*, right panel), we observed a significant increase in the current amplitude, i.e. recovery, that reached a steady-state after ~100 s (*Figure 5b*). The Cav3.3 current increased furthermore after the stopping of the simulation for 2 min (*Figure 5b*). The average current amplitude increase during these experiments was ~400% when using a 0.2 Hz TP stimulation, and up to 600% at 1 Hz (*Figure 5d*). Similarly, large effects on the Cav3.1 current were also found in these experiments (*Figure 5c*). Whereas the Cav3.1 current decreased only modestly at HP −100 mV (~10%, see also *Figure 2g*), the increase in the Cav3.1 current following the switch to HP −70 mV reached ~600% (*Figure 5c and d*) as described for the Cav3.3 current. This strong recovery of Cav3.3 and Cav3.1 currents at depolarized HPs clearly indicate that the shift in the steady-state inactivation is a dynamic and reversible mechanism. Importantly, no increase in Cav3.3 and Cav3.1 current was obtained in cells dialyzed with BAPTA (*Figure 5d*), further confirming a local $Ca^{2+}$-dependent feedback mechanism.

Interestingly, we did not observe any variation in the Cav3.2 current in these experiments (*Figure 5d* and *Figure 5—figure supplement 1b*). This finding suggests that because of its biophysical properties, i.e. rapid inactivation kinetics (as compared to Cav3.3) and its slow recovery from inactivation (as compared to Cav3.1 and Cav3.3) (*Klöckner et al., 1999*; *Kozlov et al., 1999*; *Satin and Cribbs, 2000*; *Chemin et al., 2002a*; *Perez-Reyes, 2003*), the Cav3.2 current generated in fast TP stimulation does not allow sufficient $Ca^{2+}$ entry to induce the $Ca^{2+}$-dependent regulation observed for Cav3.1 and Cav3.3 channels. To directly test this hypothesis, we have studied a Cav3.2 gain of function mutant at Met1549, recently identified in patients with hypertension due to primary aldosteronism (*Scholl et al., 2015*; *Daniil et al., 2016*). The Met1549Ile Cav3.2 mutant presents slower inactivation and deactivation kinetics and is expected to induce much larger $Ca^{2+}$ entry than the wild-type channel (*Daniil et al., 2016*). We found that the Met1549Ile Cav3.2 current decreased by ~15% during 1 Hz stimulation at HP −100 mV, whereas the current progressively increased when switched at HP −80 mV to reach a ~300% increase (*Figure 5—figure supplement 1c*). Altogether, these results demonstrate a common $Ca^{2+}$-dependent modulation mechanism for the three Cav3 currents, which depends mainly on the amount of the $Ca^{2+}$ entry and on Cav3 biophysical properties.

## $Ca^{2+}$-dependent inhibition of Cav3.3 current during action potential (AP) clamp stimulation

In order to investigate whether the $Ca^{2+}$-dependent modulation of the T-type current occurred during more physiological paradigms, we recorded Cav3.3 current during a voltage-clamp protocol mimicking thalamic neuronal activities, which was previously described in details (*Chemin et al., 2002a*). In these experiments, we found that the Cav3.3 current progressively increased during the first burst of spikes and then progressively decreased during the time course of the stimulation (*Figure 6a*). In addition, a 'rebound' in the Cav3.3 current was clearly associated with the depolarization after potential (DAP) transition, as previously described (*Chemin et al., 2002a*). We estimated the current increase as the ratio of the Cav3.3 current obtained at the fourth spike to the first one (*Figure 6b*), whereas the decrease of the current was estimated as the ratio of the Cav3.3 current obtained at the thirteenth spike to the first one (*Figure 6c*). Interestingly, the Cav3.3 current increase was similar when cells were dialyzed with an intracellular medium containing either EGTA or BAPTA

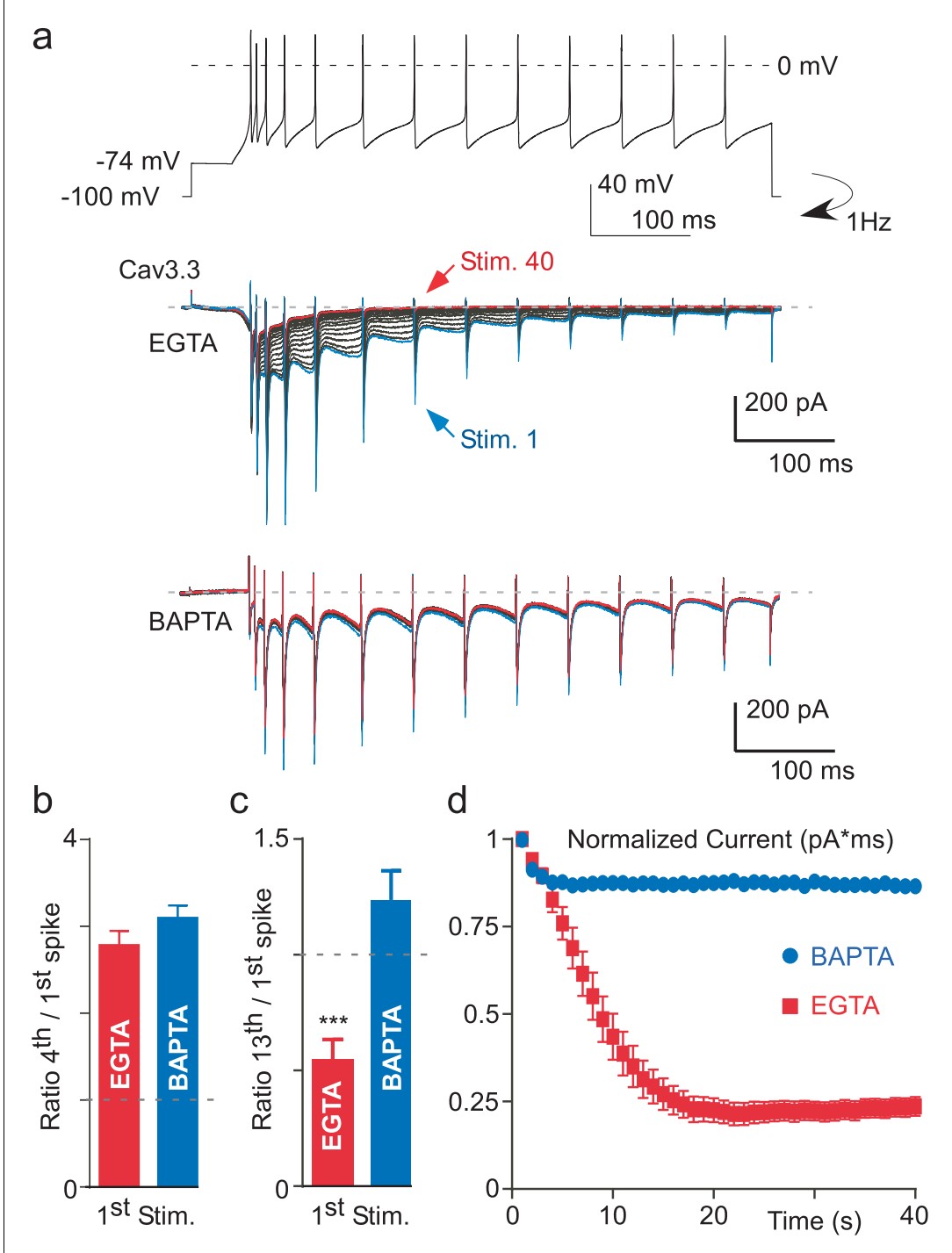

**Figure 6.** Ca$^{2+}$-dependent inhibition of the Cav3.3 current during action potential-clamp experiments. (**a**) The top trace represents a burst activity of a thalamic neuron which was used as a waveform. Typical Cav3.3 current recorded after dialyzing the cell with EGTA (middle panel) or with BAPTA (lower panel). (**b**) Average increase of the Cav3.3 current during the first spikes recorded at the first stimulation. The current increase is quantified as the ratio of the current recorded at the fourth to the first spike (n = 12–14). (**c**) Average decrease of the Cav3.3 current during the first stimulation quantified as the ratio of the thirteenth to the first spike (n = 12–14). (**d**) Time course of the total Cav3.3 current during 40 s stimulation. The total Cav3.3 current is quantified as the area under the curve (pA * ms) (n = 12–14).

and reached in average ~300%, suggesting that the current increase is not dependent of intracellular $Ca^{2+}$ (p>0.05, *Figure 6b*). In contrast, the decrease in the Cav3.3 current was bigger in cells dialyzed with EGTA as compared to BAPTA-dialyzed cells (p<0.001, *Figure 6c*). Importantly, these results were obtained during the first stimulation of the Cav3.3 current suggesting that the $Ca^{2+}$-dependent modulation of the Cav3.3 current could have a strong neuronal impact. To further investigate the behavior of the Cav3.3 current during AP clamp experiments, we performed this stimulation several times at a frequency of 1 Hz whereas the cells were clamped at HP $-100$ mV between each stimulation to allow Cav3.3 current recovery (*Figure 6a*). We found that the Cav3.3 current recorded during an AP as well as the 'rebound' in the Cav3.3 current associated with the DAP progressively decreased when the protocol was repeated 40 times in cells dialyzed with EGTA (*Figure 1a*). To account for the total Cav3.3 current variation, we calculated the integral of the Cav3.3 current at each stimulation (*Figure 6d*). This analysis revealed that the total Cav3.3 current decrease was ~80% in cells dialyzed with EGTA whereas the current decrease was less than 15% in cells dialyzed with BAPTA (p<0.001, *Figure 6d*), indicating further the robust $Ca^{2+}$-dependent modulation of Cav3.3 current during AP-clamp stimulation.

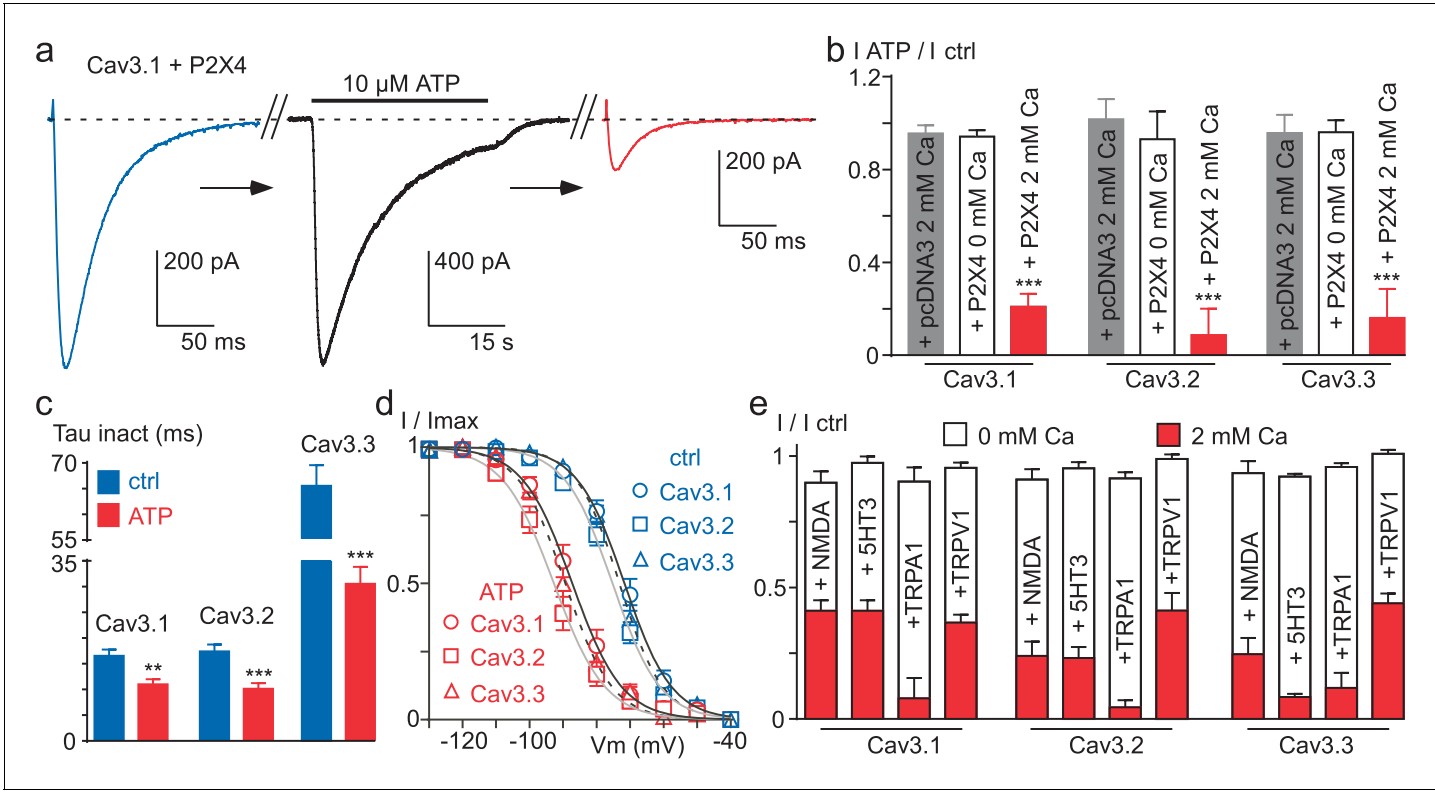

**Figure 7.** $Ca^{2+}$ entry via $Ca^{2+}$-permeable ionotropic receptors inhibits Cav3 currents. (**a**) Effect of 10 µM ATP application on the Cav3.1 current recorded in a tsA-201 cell expressing the P2X4 receptor. The Cav3.1 current recorded just before (blue trace) and just after (red trace) the application of an ATP solution (black trace, P2X4 current). (**b**) Summary of ATP effect on the Cav3.1, the Cav3.2 and the Cav3.3 current recorded in tsA-201 cells expressing or not (pcDNA3) the P2X4 receptor in the presence or in the absence of 2 mM $Ca^{2+}$ in the extracellular solution (n = 5–23 per bar). (**c**) Inactivation kinetics of the Cav3 currents in the absence and in the presence of ATP (2 mM $Ca^{2+}$, n = 11–18 per bar). (**d**) Steady-state inactivation properties of the Cav3 currents the presence and in the absence of ATP (2 mM $Ca^{2+}$, n = 10–17 per point). (**e**) Summary of the effect of NMDA, 5-HT3, TRPA1 and TRPV1 receptor activation on the Cav3 current amplitude recorded in the presence or in the absence of 2 mM $Ca^{2+}$ in the extracellular saline (n = 5–30 per bar). In all these experiments the HP is $-80$ mV.

The following figure supplement is available for figure 7:

**Figure supplement 1.** Effect of P2X4 activation on membrane density of Cav3.3 channels.

# Activation of Ca²⁺-permeable ionotropic receptors also induces Cav3 current decrease

We next investigated whether activation of a Ca²⁺-permeable ionotropic receptor might also induce Cav3 current inhibition, including Cav3.2. In cells co-expressing the purinergic P2X4 receptor and Cav3.1, the Cav3.1 current was strongly decreased after extracellular perfusion of a solution containing the purinergic agonist ATP, which generated an inward current of several seconds (*Figure 7a*). Similar results were obtained for Cav3.2 and Cav3.3 currents, and ATP caused in average ~80% inhibition of the three Cav3 current (*Figure 7b*). The decrease in Cav3 current induced by P2X4 activation was coupled to an acceleration of the current inactivation kinetics (*Figure 7c*) and a negative shift in the steady-state inactivation (*Figure 7d*). Importantly, these effects were absent when similar experiments were performed in the absence of extracellular Ca²⁺ (*Figure 7b*). Because the current decrease could involve a change in the cell surface density of the channels, we have investigated whether the membrane expression of Cav3.3 channels would be modulated by P2X4 activation (*Figure 7—figure supplement 1*). To this end, we used a Cav3.3 channel construct containing an extracellular HA tag (*Baumgart et al., 2008*), which allows the measure of its cell surface expression by enzyme-linked immunosorbent assay/luminometry (*Figure 7—figure supplement 1a*). We found that the ATP treatment did not induce significant change in membrane expression of Cav3.3-HA channels in cells co-transfected with either the P2X4 receptor or the pcDNA3 plasmid (*Figure 7—figure supplement 1b*), suggesting that the current decrease did not involve a change in the cell surface density of the channels. To extend these findings to other classes of physiologically relevant ionotropic receptors, electrophysiological experiments were performed with the Ca²⁺-permeable NMDA, 5-HT3, TRPA1 and TRPV1 receptors (*Figure 7e*). According to the results obtained with the P2X4 receptors, activation of all these receptors produced a 60% to 90% decrease of Cav3.1, Cav3.2 and Cav3.3 currents and this effect was only observed in the presence of extracellular Ca²⁺ ($p < 0.001$, *Figure 7e*). This set of experiments further demonstrate that Ca²⁺ entry into cells controls a Ca²⁺-dependent modulation of Cav3 availability.

## Discussion

This study reveals that T-type / Cav3 channels are dynamically regulated by changes in intracellular Ca²⁺ concentration. This novel regulation involves a Ca²⁺-dependent modulation of Cav3 availability. It was unraveled by demonstrating that a rise in submembrane Ca²⁺ at the vicinity of the Cav3 channels can cause a hyperpolarizing shift in the steady-state inactivation, leading to a strong Cav3 current decrease at physiological resting membrane potentials. This localized increase in intracellular Ca²⁺ can be generated by the Cav3 channel activity itself, especially for Cav3.3 channels, or by other routes of Ca²⁺ entry through the plasma membrane as obtained following activation of various Ca²⁺-permeable ionotropic receptors, all inducing a potent inhibition of Cav3.1, Cav3.2 and Cav3.3 currents. Importantly, all the effects described here were observed in 2 mM Ca²⁺-containing saline, which corresponds to the physiological range of extracellular Ca²⁺ concentration in native tissues (*Jones and Keep, 1988*), and were lost in the presence of intracellular BAPTA.

The Ca²⁺-dependent modulation revealed here was best evidenced with Cav3.3, which allows the largest Ca²⁺ entry among Cav3 channel family (*Klöckner et al., 1999*; *Kozlov et al., 1999*; *Chemin et al., 2002a*; *Perez-Reyes, 2003*). The decrease in Cav3.3 current amplitude was well correlated with the increase in the frequency of the TP stimulation. The reduction in Cav3.3 current amplitude was ~50% at 1 Hz frequency of stimulation and reached ~80% during neuronal activities in action potential clamp, suggesting that intracellular Ca²⁺ could be an important physiological regulator of Cav3 current. Importantly, this effect was very dynamic both in its induction ($T_{0.5} \sim 18$ s) and in its recovery ($T_{0.5} \sim 20$ s). Altogether, our results demonstrate that the Cav3 current decrease is linked to the cumulative Ca²⁺ entry through Cav3 channels. This Ca²⁺-dependent inhibition of Cav3 channels was observed only in the presence of external Ca²⁺ indicating that a voltage-dependent component in the mechanism is unlikely to contribute to the reported effect. Importantly, this Cav3 current inhibition induced by the Ca²⁺ entry is observed using intracellular EGTA but not anymore after the cell dialysis with BAPTA. These data reveal that this Ca²⁺-dependent modulation of Cav3 channels involves a localized increase in submembrane Ca²⁺ at the vicinity of the Cav3 channels without the need for a global Ca²⁺ increase (*Marty and Neher, 1985*; *Roberts, 1993*; *Deisseroth et al., 1996*). Interestingly, in the inside-out patch configuration, the direct application of Ca²⁺-containing

solutions up to 100 µM to the internal side of the membrane did not produce the current inhibition, suggesting that this phenomenon requires some additional components preserved in the whole-cell configuration.

A key finding of this study is that the current inhibition is closely linked to the resting membrane potential (Vm; see the schematic representation in *Figure 7*). On one hand, an increase in submembrane $Ca^{2+}$ promotes a strong negative shift in the steady-state inactivation properties of T-type channels, leading to a more prominent inhibition of the Cav3 current at physiological Vm (~ −65/−85 mV; *Figure 7*). On the other hand, the $Ca^{2+}$ entry via T-type channels is maximal at negative Vm (~ −100 mV), but the consequence of the shift in the steady-state inactivation is minimized at negative Vm (*Figure 7*). Consequently, the inhibition of the Cav3 current is maximal at physiological resting membrane potentials after stimulation at negative HP. Using this paradigm our data revealed an important finding: a wide range of modulation (~600%) of both Cav3.3 and Cav3.1 currents, but not of Cav3.2 current which was resistant to this modulation.

The lack of $Ca^{2+}$-dependent modulation of Cav3.2 channels using fast TP stimulation protocols could be explained by the specific electrophysiological properties of this Cav3 isoform. It is expected that the Cav3.2 current would generate little cumulative $Ca^{2+}$entry because of its rapid inactivation kinetics combined with its slow recovery from inactivation (*Klöckner et al., 1999*; *Kozlov et al., 1999*; *Satin and Cribbs, 2000*; *Chemin et al., 2002a*; *Perez-Reyes, 2003*). Interestingly, the Met1549Ile gain-of-function mutant of the Cav3.2 channel that displays very slow inactivation and deactivation kinetics (*Scholl et al., 2015*; *Daniil et al., 2016*), exhibits a significant $Ca^{2+}$-dependent inhibition and a wide range of modulation (~300%). Collectively, our data indicate that the $Ca^{2+}$-dependent mechanism described here can modulate all three Cav3 isoforms. Overall, the amplitude of the modulation (Cav3.3 > Cav3.1 >> Cav3.2 ~ 0) critically depends on the amount of the $Ca^{2+}$ entry and therefore relies on the biophysical properties of each Cav3 isoform.

The large $Ca^{2+}$ entry generated by the activation of ionotropic receptors induces a strong inhibition (~85%) of all three Cav3 currents, including wild-type Cav3.2. We observed this effect with a variety of $Ca^{2+}$-permeable ionotropic receptors, including the purinergic P2X4, the glutamatergic NMDA, the serotoninergic 5-HT3 and the TRP (TRPV1 and TRPA1) receptors. Notably, no inhibition of the T-type current is observed when these experiments are performed in the absence of extracellular $Ca^{2+}$, demonstrating the critical role played by the $Ca^{2+}$ entry in this mechanism of Cav3 inhibition. It is noteworthy that the tsA-201 cells, which derived from HEK-293 cells, endogenously express another class of purinergic receptors, P2Y, which are Gq-coupled receptors and induce a global increase in intracellular $Ca^{2+}$ in response to ATP (*Chemin et al., 2000*; *Fischer et al., 2005*). Interestingly, we observed no change Cav3 current properties in pcDNA3 transfected cells (*Figure 6b*) indicating that Cav3 currents are not inhibited in P2X untransfected cells following ATP application. These data suggest that activation of P2Y receptors is unable to mediate a $Ca^{2+}$-dependent modulation of Cav3 currents further supporting a membrane-delimited mechanism for the $Ca^{2+}$-dependent modulation of Cav3 channels.

Since the initial discovery of T-type currents, it was admitted that these channels were not regulated by intracellular $Ca^{2+}$ ions or by changes in intracellular $Ca^{2+}$ concentration because they do not present CDI (*Carbone and Lux, 1984*; *Fedulova et al., 1985*; *Bean, 1985*; *Bossu et al., 1985*; *Bossu and Feltz, 1986*; *Dupont et al., 1986*). More recently, these findings were confirmed using cloned T-type channels (*Staes et al., 2001*; *McRory et al., 2001*; *Perez-Reyes, 2003*) but see (*Lacinová et al., 2006*). Indeed, our data showing that the inactivation kinetics are similar at the beginning of the stimulation obtained in external $Ca^{2+}$ and in external sodium (*Figure 2e*) also support the lack of CDI for Cav3 channels. Accordingly, the important structural motifs for CDI present on L-type channels, as the $Ca^{2+}$ (EF Hand) and the calmodulin (IQ motif) binding sites are absent on the C-terminus of Cav3 channels (*Staes et al., 2001*; *McRory et al., 2001*; *Perez-Reyes, 2003*). Therefore, CDI was considered in pioneer studies as a hallmark to distinguish between L-type and T-type $Ca^{2+}$ currents (*Carbone and Lux, 1984*; *Fedulova et al., 1985*; *Bean, 1985*; *Bossu et al., 1985*; *Bossu and Feltz, 1986*; *Dupont et al., 1986*). Increasing the intracellular $Ca^{2+}$ concentration $[Ca^{2+}]_{INT}$ from $10^{-10}$ to $10^{-7}$ M or even to $10^{-6}$ M was classically used to isolate a 'pure' T-type current, which presented no change in its inactivation kinetics, whereas the L-type current disappeared because of the acceleration of its 'run-down' (*Bossu et al., 1985*; *Bossu and Feltz, 1986*; *Dupont et al., 1986*). Interestingly, in these seminal studies, the native T-type current was mostly related to Cav3.2 (nickel-sensitive) channels (*Carbone and Lux, 1984*; *Fedulova et al., 1985*;

*Bean, 1985*; *Bossu et al., 1985*; *Bossu and Feltz, 1986*; *Dupont et al., 1986*), for which we show that the intrinsic electrophysiological properties do not allow the triggering of the $Ca^{2+}$-dependent current inhibition, contrary to Cav3.1 and Cav3.3 channels. However, and consistent with our present findings, these early studies performed in the presence of an increasing amount of $[Ca^{2+}]_{INT}$ have reported important features of the T-type current: (1) at $10^{-8}$ M, the T-type current was stable during 60 min whereas at $10^{-7}$ M the T-type current decreased and was suppressed after 10–15 min (*Bossu et al., 1985*); (2) at $10^{-7}$ M, an hyperpolarizing shift (~10 mV) of the steady-state inactivation occurred (*Bossu and Feltz, 1986*); and (3) at $10^{-6}$ M no T-type current was recorded at HPs above $-100$ mV (*Dupont et al., 1986*). Altogether, these historical results and our present data are in favor of a 'minute scale' $Ca^{2+}$-dependent modulation of the T-type current, which induces a hyperpolarizing shift in the steady-state inactivation and consequently a decrease in T-type current amplitude (see *Figure 7*).

It is also important to depict our results in the light of recent findings obtained on L-type channels using high frequency of stimulation (*Oliveria et al., 2007*, *2012*). In 15 mM external $Ca^{2+}$, the Cav1.2 current dropped to 40% of the control amplitude during 1 Hz stimulation and this inhibition was abolished in the presence of intracellular BAPTA (*Oliveria et al., 2007*, *2012*). The Cav1.2 current decrease was stable after 3–5 min, similar to that obtained with 40–50 s stimulation for Cav3.3 in the presence of the physiological 2 mM $Ca^{2+}$ concentration, and the recovery was almost total after 4–5 min. However, contrary to Cav3 currents, the decrease in the Cav1.2 current stimulated at 1 Hz did not involve a shift in its steady-state inactivation (*Oliveria et al., 2007*, *2012*), suggesting distinct mechanisms for the L-type and the T-type current modulation. In addition, a prolonged stimulation of Cav1.2 current (in 10 mM external $Ca^{2+}$ saline) induced channel endocytosis (*Green et al., 2007*; *Tsuruta et al., 2009*; *Hall et al., 2013*) and, in this case, the recovery of the L-type current took approximately 30 min (*Green et al., 2007*). This $Ca^{2+}$-dependent modulation of L-type channels appears distinct of that described here for Cav3 channels, both regarding the time of the recovery and the shift in the steady-state inactivation. In addition, we found that ATP treatment in cells expressing both the P2X4 receptors and the Cav3.3 channels did not induce significant changes in membrane expression of Cav3.3, suggesting that the current decrease did not involve an endocytosis mechanism. Therefore, although the L- and the T-type channel regulation share apparent similar properties, modulation of the T-type current by cumulative $Ca^{2+}$ entry has unique features, depending mostly on the shift of the steady-state inactivation, i.e. the modulation of Cav3 availability.

The discovery of a $Ca^{2+}$-driven feedback regulation of T-type channels may have important physiological and pathophysiological implications. Indeed, an increase in the activity of T-type channels have been implicated in several diseases linked to altered $Ca^{2+}$ signaling (*Orestes et al., 2013*; *Jagodic et al., 2007*; *Scholl et al., 2015*; *Zamponi, 2016*; *Daniil et al., 2016*) and T-type channel activity is also linked to $Ca^{2+}$ overload toxicity occurring in ischemia (*Nikonenko et al., 2005*; *Bancila et al., 2011*; *Gouriou et al., 2013*). Also, our study reveals that activation of $Ca^{2+}$-permeable ionotropic receptors could also markedly inhibit T-type channel activity, and interestingly, cross-talk between these receptors and T-type channels have been recently observed (*Comunanza et al., 2011*; *Kerckhove et al., 2014*; *Wang et al., 2015*; *Tatsuki et al., 2016*).

In summary, we have identified a novel regulation pathway for T-type $Ca^{2+}$ channels. By demonstrating that $Ca^{2+}$ entry exerts a feedback control on T-type channel activity, our study opens up new horizons towards deciphering how this local and dynamic $Ca^{2+}$-dependent modulation of Cav3 channels can impact the cellular and physiological roles of T-type channels in normal and disease states.

## Materials and methods

### Cell culture and transfection protocols

tsA-201 cells (RRID:CVCL_2737) were obtained from the European Collection of Authenticated Cell Cultures (ECACC 96121229). The identity of tsA201 has been confirmed by STR profiling and the cells have been eradicated from mycoplasma at ECACC. We routinely tested the cells for the absence of the mycoplasma contamination. Cells were cultivated in DMEM supplemented with GlutaMax, 10% fetal bovine serum and 1% penicillin / streptomycin (Invitrogen, Fisher Scientific, France). Transfections were performed using jet-PEI (Ozyme, France) with a DNA mix (1.5 μg total) containing

0.5% of a GFP encoding plasmid and 99.5% of either of the plasmids (pcDNA3.1) that code for the human Cav3.1a, Cav3.2, Cav3.3 and Met1549Ile Cav3.2 constructs. In experiments with ionotropic receptors, 1 µg of either of the plasmid constructs that code for human P2X4, mouse 5-HT3, human TRPV1, mouse TRPA1 and rat NMDA receptor (NR1A and NR2A (0.5 µg each)) were added to the DNA mix. Two days after transfection, tsA-201 cells were dissociated with Versene (Invitrogen, Fisher Scientific, France) and plated at a density of ~35 × 10³ cells per 35 mm Petri dish for electrophysiological recordings, which were performed the following day.

## DRG neurons

All animal use procedures were done in accordance with the directives of the French Ministry of Agriculture (A 34-172-41). Dorsal root ganglion (DRG) neurons were prepared as described earlier (*Voisin et al., 2016*). Briefly, adult male C57BL/6J mice were anaesthetized with pentobarbital injection and transcardially perfused with HBSS (pH 7.4, 4°C). Lumbar DRGs with attached roots were dissected and collected in cold HBSS supplemented with 5 mM HEPES, 10 mM D-glucose and 1% penicillin/streptomycin. DRGs were treated with 2 mg/ml collagenase II and 5 mg/ml dispase for 40 min at 37°C, washed in HBSS and resuspended in 1 ml of neurobasal A medium supplemented with B27, 2 mM L-glutamine and 1% penicillin/streptomycin (Invitrogen, Fisher Scientific, France). Single-cell suspensions were obtained by 5 passages through three needle tips of decreasing diameter (gauge 18, 21, and 26). Cells were plated onto polyornithine/laminin-coated dishes. After 2 hr, the medium was removed and replaced with neurobasal B27 supplemented with 10 ng/ml neurotrophin 4 (NT4) and 2 ng/ml glial derived neurotrophic factor (GDNF). Patch clamp recordings were performed 6–24 hr after plating on neurons with a 'rosette' morphology corresponding to D-hair neurons that express alarge density of T-type current (*Dubreuil et al., 2004*; *Voisin et al., 2016*).

## Electrophysiological recordings

Macroscopic currents were recorded at room temperature using an Axopatch 200B amplifier (Molecular Devices, Sunnyvale CA). Borosilicate glass pipettes had a resistance of 1.5–2.5 MOhm when filled with an internal solution containing (in mM): 140 CsCl, 10 EGTA, 10 HEPES, 3 Mg-ATP, 0.6 GTPNa, and 3 CaCl$_2$ (pH adjusted to 7.25 with KOH, ~315 mOsm, ~100 nM free Ca$^{2+}$ using the Max-Chelator software, http://maxchelator.stanford.edu/). Similar results were obtained using either 10 mM or 20 mM EGTA. In some experiments, BAPTA (20 mM) was substituted with EGTA. The extracellular solution contained (in mM): 135 NaCl, 20 TEACl, 2 CaCl$_2$, 1 MgCl$_2$, and 10 HEPES (pH adjusted to 7.25 with KOH, ~330 mOsm). To avoid inhibition of 5-HT3 and NMDA receptors, NaCl was substituted with TEACl and MgCl$_2$ was also omitted in NMDA experiments. In the cell-free inside-out patch experiments the intrapipette solution contained 100 mM BaCl$_2$ and 10 mM HEPES (pH adjusted to 7.25 with NaOH, ~310 mOsm) and the bath solution contained (in mM): 145 KCl, 10 HEPES and 1 MgCl$_2$ (pH adjusted to 7.25 with KOH, ~305 mOsm). In the inside-out configuration the patch was perfused with the bath solution supplemented with either 1 mM EGTA or increasing the concentration of CaCl$_2$ (1, 10 and 100 µM). For D-hair neuron recordings, the bath solution contained (in mM): 140 NaCl, 10 HEPES, 5 KCl, 2 CaCl$_2$, 1 MgCl$_2$ and 10 glucose (pH adjusted to 7.25 with NaOH, ~330 mOsm) and cells were perfused with an extracellular solution containing (in mM): 140 TEACl, 10 HEPES, 5 KCl, 2 NaCl, 2 CaCl$_2$, 1 MgCl$_2$ and 10 glucose (pH adjusted to 7.25 with TEAOH, ~330 mOsm). Recordings were filtered at 2 kHz. Steady-state inactivation curves were fitted using the Boltzmann equation where I/I max = $1/(1+\exp((V_m-V_{0.5})/\text{slope factor}))$. Data were analyzed using pCLAMP9 (Molecular Devices) and GraphPad Prism (GraphPad) softwares. Results are presented as the mean ± SEM, and n is the number of cells. Statistical analysis was performed with the Student *t*-test or with one-way ANOVA combined with a Tukey post-test for multiple comparisons (*p<0.05, **p<0.01, ***p<0.001).

## Modelling

Modelling was performed using the NEURON simulation environment (*Hines and Carnevale, 1997*). The model was modified to simulate voltage-clamp experiments in thalamic reticular neurons (*Destexhe et al., 1996*) (as available from the model database at Yale University (https://senselab.med.yale.edu/modeldb/). The electrophysiological properties of the Cav3.3 channels were modelled using Hodgkin-Huxley equations as described previously (*Huguenard and Prince, 1992*;

*Destexhe et al., 1996*). The values obtained for Cav3.3 were substituted for the corresponding values of native T-channels in thalamic reticular neurons (*Huguenard and Prince, 1992*), as previously described (*Chemin et al., 2002a*). To match the voltage clamp data, the modelling experiments were performed at 24°C.

The equations to model the Cav3.3 current properties at rest were:

$$
\begin{aligned}
m\infty \quad &= 1/(1+\exp(-(v+52.6)/4.7)) \\
h\infty \quad &= 1/(1+\exp((v+72.4)/5.7)) \\
tau_m \quad &= (1.377+1.512/(\exp((v+12.52)/14.38)+\exp(-(v+81.59)/5))) \\
tau_h \quad &= (65.34+1/(\exp((v+41.04)/4.01)+\exp(-(v+333.1)/46.86)))
\end{aligned}
$$

The equations to model the Cav3.3 current properties after 40 s stimulation at 1 Hz frequency were:

$$
\begin{aligned}
m\infty \quad &= 1/(1+\exp(-(v+55.9)/5.6)) \\
h\infty \quad &= 1/(1+\exp((v+81.1)/5)) \\
tau_m \quad &= (1.141+0.9592/(\exp((v+14.95)/13.97)+\exp(-(v+81.53)/5))) \\
tau_h \quad &= (26.55+0.66/(\exp((v+32.42)/6.4)+\exp(-(v+225)/22.21)))
\end{aligned}
$$

## Luminometric analysis of HA-tagged Cav3.3 channels

The membrane expression of the Cav3.3 channel was quantified as previously described (*Chemin et al., 2007*). The tsA-201 cells were cultured in 24-well plates and co-transfected with a Cav3.3-HA construct (*Baumgart et al., 2008*) and either the P2X4 receptor or the pcDNA3 plasmid (ratio 1:1). Two days after transfection, ATP treatments were performed as in the electrophysiological experiments. The cells were washed with the electrophysiological extracellular solution containing 3 µM ivermectin and afterward 10 µM ATP was applied for 30–45 s at room temperature. Then the cells were directly fixed for 5 min in PBS containing 4% paraformaldehyde. After three PBS washes, the cells were incubated for 30 min in blocking solution (PBS plus 1% fetal bovine serum). The Cav3.3-HA protein was detected using a rat anti-HA monoclonal antibody (1:1000 dilution; clone 3F10, Roche Diagnostics, France) after incubation for 1 hr at room temperature. After four washes with PBS plus 1% fetal bovine serum for 10 min, cells were incubated for 30 min with horseradish peroxidase-conjugated goat anti-rat secondary antibody (1:1000 dilution; Jackson ImmunoResearch Laboratories, West Grove, PA). Cells were rinsed four times with PBS for 10 min before addition of SuperSignal enzyme-linked immunosorbent assay Femto maximum sensitivity substrate (Pierce, Fisher Scientific, France). Luminescence was measured using a VICTOR2 luminometer (PerkinElmer Life Sciences, Waltham, MA), and the protein amount in each well was then measured using the BCA assay (Pierce, Fisher Scientific, France) to normalize the measurements. All data were normalized to the level of signal obtained in P2X4 transfected cells without the ATP treatment. Each experiment was performed in quadruplicate and three independent sets of transfection experiments were performed under each condition. The results are presented as the mean ± SEM.

## Chemical reagents

Compounds were purchased from Sigma (France). To activate ionotropic receptors, we used 10 µM ATP in the presence of 3 µM ivermectin for P2X4, 100 µM glutamate in the presence of 100 µM glycine for NMDA, 10 µM serotonin for 5-HT3, 0.5 µM capsaicin for TRPV1 and 100 µM allyl isothiocyanate for TRPA1. Drugs were applied using a gravity-driven homemade perfusion device and control experiments were performed similarly using the vehicle alone.

## Acknowledgements

We thank Drs FA Rassendren for the P2X4 and the TRPV1, E Boué-Grabot for the 5-HT3, K Talavera for the TRPA1 and H Hirbec for the NR1A and NR2A plasmid constructs. We are grateful to Dr. T Voisin for DRG neurons culture and Drs Nathalie C Guerineau and Steve Dubel for critical reading of the manuscript.

## Additional information

### Funding

| Funder | Grant reference number | Author |
|---|---|---|
| Agence Nationale de la Recherche | ANR-10-BLAN-1601 | Philippe Lory |
| Laboratory of excellence in Ion Channel Science and Therapeutics | LabEx ICST | Philippe Lory |

The funders had no role in study design, data collection and interpretation, or the decision to submit the work for publication.

### Author contributions

MC, Formal analysis, Investigation; IB, Investigation; PL, Writing—review and editing; JC, Conceptualization, Formal analysis, Supervision, Investigation, Methodology, Writing—original draft, Writing—review and editing

### Author ORCIDs

Jean Chemin, http://orcid.org/0000-0002-6089-5964

### Ethics

Animal experimentation: All animal use procedures were done in accordance with the directives of the French Ministry of Agriculture (A 34-172-41).

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
