## [Decision Letter]

Thank you for submitting your article "Activity-dependent regulation of T-type calcium channels by submembrane calcium ions" for consideration by *eLife*. Your article has been favorably evaluated by Richard Aldrich as the Senior Editor and three reviewers, one of whom, Baron Chanda (Reviewer #1), is a member of our Board of Reviewing Editors. The following individual involved in review of your submission has agreed to reveal their identity: Henry M. Colecraft (Reviewer #3).

The reviewers have discussed the reviews with one another and the Reviewing Editor has drafted this decision to help you prepare a revised submission. We hope you will be able to submit the revised version within two months.

Summary:

The reviewers agree that this is an interesting study that convincingly establishes that calcium ions are able to alter the function of native and transiently expressed T-type calcium channels. They show that calcium entry via the channel itself is able to induce this type of feedback inhibition, similar to what has been observed in Cav1.x and Cav2.x channels. The notion that intracellular calcium regulates T-type channels is not accepted or highlighted in the T-type literature. The T-type channels also lack the Ca^2+^ and calmodulin binding motifs found in L-type channels. In that context, this work is important because it provides the clearest evidence to date of Ca^2+^ regulation of distinct T-type channel isoforms in a manner that could potentially be physiologically important. The study also highlights the conserved nature of calcium regulation in voltage-gated calcium channels.

Essential Revisions:

Despite overall enthusiasm about this study, there were number of concerns that should be addressed in the revised version.

1) The magnitude of the inhibition of T-type current seen (> 80%) in Figure 6 appear much larger than can be accounted for by the sole mechanism proposed by the authors; i.e. a 5-10 mV hyperpolarizing shift in the steady-state inactivation curves. It is possible there is also a rightward shift in the voltage-dependence of the activation curve that contributes to the apparent inhibition seen. The authors should determine whether the activation curve is shifted under one of the conditions in which the steady-state inactivation is left-shifted.

Alternatively, this can be addressed by building quantitative models to show whether calcium dependent shifts in inactivation can account for this inhibition.

2) The electrophysiological pulse protocols used to elicit Ca^2+^-dependent inhibition of the currents are fairly non-physiological. This is fine for probing the phenomenon and the underlying biophysical mechanisms. Nevertheless, the authors speculate about the physiological implications of their results. The impact of the study would be enhanced if the phenomenon could be demonstrated (perhaps in Cav3.3 where the effect is most robust) with more physiological activation paradigms such as action potential waveforms. Alternatively, I think what would be more compelling is to show that under physiological conditions (i.e., in neurons), native calcium permeable receptors are able to modulate native T-type currents, and by extension, affect neuronal output.

3) A previous study (Lacinova et al., 2006, not cited by the authors) has reported calcium dependent effects on Cav3.1 channel gating that are opposite to those reported here and a similar calcium dependent slowing of inactivation kinetics of Cav3.1 was reported by Monteil et al., 2000 (see Figure 6 in that study). This should be discussed and discrepancies to the present work explained.

4) Please discuss how big is the shift in half inactivation voltage after ionomycin treatment – i.e., is it greater than what is observed upon repetitive stimulation?

5) We realize that the trying to figure out the molecular mechanism of modulation is not within the scope of this manuscript, we would like to suggest that the authors add some simple experiments that would shed light on the mechanism. For instance, is this inhibition observed in inside out patches. Because if it is not a direct effect and requires accessory proteins, one would expect that this inhibition will not be observed. We believe that this should be a straightforward and insightful experiment.

---

## [Author Response]

*Essential Revisions:*

*Despite overall enthusiasm about this study, there were number of concerns that should be addressed in the revised version.*

1) The magnitude of the inhibition of T-type current seen (> 80%) in Figure 6 appear much larger than can be accounted for by the sole mechanism proposed by the authors; i.e. a 5-10 mV hyperpolarizing shift in the steady-state inactivation curves. It is possible there is also a rightward shift in the voltage-dependence of the activation curve that contributes to the apparent inhibition seen. The authors should determine whether the activation curve is shifted under one of the conditions in which the steady-state inactivation is left-shifted.

*Alternatively, this can be addressed by building quantitative models to show whether calcium dependent shifts in inactivation can account for this inhibition.*

We agree that in the previous version of the manuscript, we have not addressed the possibility that a shift in the voltage-dependence of the activation curve may contribute to the apparent inhibition seen. Therefore, we have performed new experiments to address this point. As the calcium-dependent current inhibition is a very dynamic process in both its induction and recovery (T_0.5_ ~ 20s) we have used a pair-pulse protocol, as presented in new version of the Figure 3. We found that the calcium-dependent inhibition is associated with a slight leftward shift (~3 mV) in the steady-state activation curve of Cav3.3 but not with a rightward shift (Figure 3 and subsection “Ca^2+^ entry during fast TP stimulation induces a negative shift in the steady-state inactivation”, first paragraph). This leftward shift in the steady-state activation is not expected to induce Cav3.3 current inhibition. In addition, we have found no significant variation in the steady-state activation curve of Cav3.1, for which the steady-state inactivation curve is significantly shifted to the left (Figure 3—figure supplement 1 and the last paragraph of the aforementioned subsection).

We have also performed a simulation of Cav3.3 properties before and after the calcium-dependent current inhibition using the NEURON simulation environment (Figure 4 and described in Methods, subsection “Modelling”). This simulation predicts that the calcium-dependent current inhibition would induce ~75% inhibition of the Cav3.3 current recorded from a holding potential of -70 mV (Figure 4 and subsection “Ca^2+^-dependent inhibition of T-type channels is higher at physiological resting membrane potentials”, first paragraph). It is important to note that this value corresponds well with those obtained during the experiments presented in Figure 5 obtained at 1 Hz frequency of stimulation.

In addition, the magnitude of the shift in the steady-state inactivation curve is higher after ionomycin treatment ~22 mV (Figure 1, see the major point 4) and after P2X4 activation (~16 mV, new Figure 7), which could explain the strong inhibition of the Cav3 current at physiological resting membrane potentials.

*2) The electrophysiological pulse protocols used to elicit Ca^2+^-dependent inhibition of the currents are fairly non-physiological. This is fine for probing the phenomenon and the underlying biophysical mechanisms. Nevertheless, the authors speculate about the physiological implications of their results. The impact of the study would be enhanced if the phenomenon could be demonstrated (perhaps in Cav3.3 where the effect is most robust) with more physiological activation paradigms such as action potential waveforms. Alternatively, I think what would be more compelling is to show that under physiological conditions (i.e., in neurons), native calcium permeable receptors are able to modulate native T-type currents, and by extension, affect neuronal output.*

We completely agree with this point and we have performed new experiments with Cav3.3 using action potential waveforms (new Figure 6). We used typical thalamic neuronal activities that we have previously used to characterize difference in behavior among Cav3 isoforms (Chemin et al., 2002 “Specific contribution of human T-type calcium channel isotypes (a1G, a1H and a1I) to neuronal excitability”). We have found that the Cav3.3 current strongly decreases during this neuronal activity (new Figure 6, subsection “Ca^2+^-dependent inhibition of Cav3.3 current during action potential (AP) clamp stimulation.”). In average the Cav3.3 current decrease was ~50% for the first stimulation, which lasted only 500 ms (Figure 6) and increases to ~ 80% at the 20^th^ stimulation (Figure 6). Importantly, no such decrease in Cav3.3 current was observed in cells dialyzed with BAPTA (Figure 6). We thank the reviewers for their suggestion, which greatly enhance the impact of our study.

*3) A previous study (Lacinova et al., 2006, not cited by the authors) has reported calcium dependent effects on Cav3.1 channel gating that are opposite to those reported here and a similar calcium dependent slowing of inactivation kinetics of Cav3.1 was reported by Monteil et al., 2000 (see Figure 6 in that study). This should be discussed and discrepancies to the present work explained.*

In the Figure 6 of our precedent article “Monteil et al., 2000”, the Cav3.1 (named α1G at this time) current was recorded at low frequencies of stimulation (0.2 Hz or 0.1 Hz) and the permeation properties of the Cav3.1 channel was tested with calcium, barium and strontium. It should be noted that at these low frequencies calcium does not induce Cav3.1 current inhibition (Figure 2 in our present article) and accordingly no inhibitory effect of calcium was found in Monteil et al., 2000. In addition, the effect of barium or strontium at low frequency of stimulation mainly depends of the Cav3 isoform used and is expected to depend on difference in residues present in (or surrounding) the P-loop of the channels (for example see McRory et al., 2001 “Molecular and functional characterization of a family of rat brain T-type calcium channels.”). Importantly, we have not described in our present study the effect of barium ions (Figure 6 in Monteil et al., 2000) but instead we have used intracellular BAPTA (as compared to EGTA) to probe calcium-dependent effect. Therefore, we believe that there is no discrepancy between our present results and those reported by Monteil et al., 2000.

In the study of Lacinova et al., 2006 (cited in the sixth paragraph of the Discussion), the Cav3.1 current was compared between different extracellular and intracellular solution. The authors mainly compare 2 extreme conditions: “high intracellular calcium concentration” which was 20 mM external calcium combined with 200 µM free internal calcium; and “0 divalent cation” which was 135 mM external sodium combined with 4 mM intracellular EGTA. Although this study presents interesting results it is very difficult to compare these results with our study (physiological external 2 mM calcium and an internal medium containing EGTA combined with calcium to match physiological concentration at rest ~100 nM) for the following reasons. First, as directly mentioned by the authors (p252), “non-specific voltage shifts due to altered surface charge took place” when comparing both conditions and therefore a shift in the steady-state inactivation (or activation) could not be quantitatively compared (as discussed p254 and p 256). Therefore, the rightward shift in the inactivation curve observed with the “high intracellular calcium concentration” was completely expected by a screen charge effect (20 mM external calcium versus 0 external divalent). Second, the dialysis of the cell for several minutes in 4 mM EGTA/0 Ca or in 200 µM Ca/ 0 EGTA may lead to major change in the cell metabolism, as the prolonged activation/inhibition of several calcium-dependent enzymes (that possibly may lead to Cav3 channel modulation), which was not the case in our study mainly comparing the effect of internal EGTA and BAPTA. However, we do not think that this study reported opposite results as compared to our study. Again, it is important to note that experiments in Lacinova et al., were performed at the frequency of stimulation of 0.2 Hz (p252), a frequency for which we found no calcium-dependent inhibition for Cav3.1 current (and therefore no shift in the steady-state inactivation, as observed in Figure 3 in Lacinova et al.). In addition, using “intermediate” solutions (that limit altered surface charge change), the authors showed that the Cav3.1 current is increased in the presence of intracellular EGTA as compared to 200 µM intracellular calcium (both in 20 mM external calcium). The same results were observed when comparing the 135 mM external sodium and the 2 mM external calcium (both with EGTA in the intracellular calcium). This suggests that an increase in intracellular calcium promotes the current inhibition (Figure 2 in Lacinova et al.). Also, in the presence of 20 mM external calcium, the inactivation kinetics (and – to a lesser extent – the activation kinetics) of the Cav3.1 current were faster in the presence of internal calcium as compared to internal EGTA (Figure 2 in Lacinova et al.). Overall, although it is very difficult to directly compare our results with those of Lacinova et al. (especially concerning the inactivation curves), we do believe that our study provides data that are not opposite or incompatible with those of Lacinova et al. (2006).

*4) Please discuss how big is the shift in half inactivation voltage after ionomycin treatment – i.e., is it greater than what is observed upon repetitive stimulation?*

We now report the voltage shift in the inactivation curve after ionomycin treatment in the new version of Figure 1. We found in sensory neurons that ionomycin induced ~22 mV shift in the half inactivation voltage of the native T-current (Figure 1 and subsection “Inhibition of the native T-type and the recombinant Cav3 currents by ionomycin”). Similarly, the ionomycin treatment induced ~23 mV leftward shift in the steady-state activation of recombinant Cav3.1, Cav3.2 and Cav3.3. This shift is 2 fold bigger than those observed upon repetitive stimulation (~10 mV, Figure 3dD and subsection “Ca^2+^ entry during fast TP stimulation induces a negative shift in the steady-state inactivation”, first paragraph) and could account for the strong inhibitory effect of ionomycin at physiological resting membrane potential (see major point 1).

*5) We realize that the trying to figure out the molecular mechanism of modulation is not within the scope of this manuscript, we would like to suggest that the authors add some simple experiments that would shed light on the mechanism. For instance, is this inhibition observed in inside out patches. Because if it is not a direct effect and requires accessory proteins, one would expect that this inhibition will not be observed. We believe that this should be a straightforward and insightful experiment.*

As proposed by the reviewers, we have performed inside-out experiments with Cav3.3 (new Figure 2—figure supplement 1). We have recorded the Cav3.3 current in the presence of 100 mM barium in the patch-pipette (in order to record significant Cav3.3 current and not to produce the possible calcium-dependent inhibition due to the permeation of external calcium, as observed in whole-cell experiments) and we have stimulated the channels with voltage-ramps applied at a frequency of 0.2 Hz (Figure 2—figure supplement 1). It is important to note that the used of external 100 mM divalent concentration induces a strong positive shift in V0.5 values due to altered surface charge (as compared to physiological 2 mM calcium, see point 3). Indeed, in the inside-out configuration, we found that at a holding potential (HP) of -50 mV, only the half of channels are inactivated (~47% ). Therefore, as the calcium effect observed in the whole-cell configuration is more robust at depolarized HPs (-70 / -80 mV), we used HP -50 mV in the inside-out configuration to match whole-cell experiments and possibly maximize the putative direct effect of calcium. Before patch excision, the cells were perfused with a solution containing 1 mM EGTA / 0 Ca and the inside-out patch configuration was performed (Figure 2—figure supplement 1). Afterward, the effect on the Cav3.3 current of calcium containing solutions (1, 10 and 100 µM) were tested and compared to the 1 mM EGTA / 0 Ca solution (Figure 2—figure supplement 1). In average we found no significant effect of these calcium solutions (Figure 2—figure supplement 1, subsection “Inhibition of Cav3.3 current at high frequency of stimulation is caused by the Ca^2+^ entry”, third paragraph) suggesting that intracellular calcium does not directly affect the Cav3.3 current. Higher concentration of calcium (mM) could not be investigated because we observed a strong tendency of the patches to form vesicles, which were very resistant to air and most of the time irreversible.